# Interaction of MRPL9 and GGCT Promotes Cell Proliferation and Migration by Activating the MAPK/ERK Pathway in Papillary Thyroid Cancer

**DOI:** 10.3390/ijms231911989

**Published:** 2022-10-09

**Authors:** Hui-Min Zhang, Zi-Yi Li, Zhou-Tong Dai, Jun Wang, Le-Wei Li, Qi-Bei Zong, Jia-Peng Li, Tong-Cun Zhang, Xing-Hua Liao

**Affiliations:** 1Institute of Biology and Medicine, College of Life and Health Sciences, Wuhan University of Science and Technology, Wuhan 430070, China; 2Department of Obstetrics and Gynecology, Tongji Hospital, Tongji Medical College, Huazhong University of Science and Technology, Wuhan 430030, China

**Keywords:** MRPL9, GGCT, Papillary Thyroid Cancer (PTC), proliferation, migration

## Abstract

Thyroid cancer remains the most common endocrine malignancy worldwide, and its incidence has steadily increased over the past four years. Papillary Thyroid Cancer (PTC) is the most common differentiated thyroid cancer, accounting for 80–85% of all thyroid cancers. Mitochondrial proteins (MRPs) are an important part of the structural and functional integrity of the mitochondrial ribosomal complex. It has been reported that MRPL9 is highly expressed in liver cancer and promotes cell proliferation and migration, but it has not been reported in PTC. In the present study we found that MRPL9 was highly expressed in PTC tissues and cell lines, and lentivirus-mediated overexpression of MRPL9 promoted the proliferation and migration ability of PTC cells, whereas knockdown of MRPL9 had the opposite effect. The interaction between MRPL9 and GGCT (γ-glutamylcyclotransferase) was found by immunofluorescence and co-immunoprecipitation experiments (Co-IP). In addition, GGCT is highly expressed in PTC tissues and cell lines, and knockdown of GGCT/MRPL9 in vivo inhibited the growth of subcutaneous xenografts in nude mice and inhibited the formation of lung metastases. Mechanistically, we found that knockdown of GGCT/MRPL9 inhibited the MAPK/ERK signaling pathway. In conclusion, our study found that the interaction of GGCT and MRPL9 modulates the MAPK/ERK pathway, affecting the proliferation and migration of PTC cells. Therefore, GGCT/MRPL9 may serve as a potential biomarker for PTC monitoring and PTC treatment.

## 1. Introduction

Thyroid cancer (TC) is currently the most common endocrine system malignancy [1]. The incidence of thyroid cancer has been on the rise worldwide over the past few decades [2,3]. The most common histological type of TC is papillary thyroid carcinoma (PTC), which accounts for approximately 80–90% of all TC cases [4,5]. Although the vast majority of PTC patients have a good prognosis with conventional treatment, recurrence and distant metastasis occur in approximately 30% of patients, thus significantly reducing survival [6,7]. In addition, the occurrence and progression of PTC are influenced by multiple factors, such as genetic mutations, environmental exposures, and epigenetic alterations [8]. Therefore, new molecular biomarkers should be discovered and used as useful tools for diagnosis and treatment management to better characterize the malignancy and aggressiveness of lesions, while providing theoretical support for the search for new PTC diagnostic and therapeutic targets.

Mitochondrial proteins (MRPs) are essential components of the structural and functional integrity of the mitochondrial ribosomal complex [9]. In mammals, more than 80 *Mrp* genes have been identified, and this group of genes is divided into two broad categories: *Mrpl*, a component of the large subunit, and *Mrps*, a component of the small subunit [10]. *MRPL9* genes are a component of the mitochondrial ribosomal large subunit, which mediates translation in mitochondria [11]. It has been reported that MRPL9 is highly expressed in liver cancer and is associated with poor prognosis [12,13], and knockdown of MRPL9 inhibits the proliferation and migration of liver cancer cells [13]. Additionally, loss-of-function of MRPL9 inhibited the colony-forming unit (CFU) potential of MDA-MB-231 and BT-549 TNBC models and increased their sensitivity to paclitaxel [14]. It can be seen that MRPL9 may be related to the proliferation and migration ability of cancer cells. However, the function of MRPL9 in PTC is unknown.

GGCT (γ-glutamylcyclotransferase) is one of the major enzymes in glutathione metabolism [15,16], catalyzing the reaction to generate 5-oxoproline and free amino acids from γ-glutamyl peptide. Oakley et al., cloned a cDNA encoding human GGCT and found that GGCT is identical to the putative protein chromosome 7 open reading frame 24 (C7orf24), which was previously registered as a putative open reading on the chromosome 7 (7p15-14) frame [17]. Before Oakley’s report, C7orf24 was known as a cancer-associated protein. Xu et al., identified 46 common cancer signature genes from a pooled DNA array database of previously reported human cancers and reported that one of the highly expressed genes was C7orf24 [18]. In addition, Kageyama et al. also characterized C7orf24 as an up-regulated protein in urothelial carcinoma specimens by proteomic analysis [19,20]. Studies have shown that GGCT is highly expressed in tumorous breast tissue [21], and that patients with high GGCT expression have a poor prognosis. In addition, GGCT is highly expressed in colorectal cancer [22], gastric cancer [23], prostate cancer [24], and human glioma [25]. Our previous study showed that GGCT is highly expressed in PTC, and knockdown of GGCT inhibited the migration ability of PTC cells [26]. In this paper, we further explore the molecular mechanism of GGCT regulating the proliferation and migration of PTC cells on the basis of previous research, and provide theoretical support for the treatment of PTC.

In this study, we evaluated MRPL9 expression levels at the clinical and cellular levels and the effects of MRPL9 deletion or enhancement on the biological behavior of PTC cells. In addition, the interaction between MRPL9 and GGCT was demonstrated, and the regulatory effect of MRPL9/GGCT on the MAPK/ERK signaling pathway was further explored. Finally, the effects of MRPL9 and GGCT on tumor growth and metastasis were further investigated in vivo.

## 2. Results

### 2.1. MRPL9 Is Highly Expressed in PTC

TCGA database analysis found that MRPL9 was highly expressed in PTC (Figure 1A). In addition, we found that the expression level of MRPL9 was associated with the prognosis of thyroid cancer, and patients with high MRPL9 expression showed poorer overall survival (http://gepia2.cancer-pku.cn/#index, accessed on 21 March 2022) (Figure 1B). To further explore the expression level of MRPL9 in PTC, 26 pairs of cancer tissues and adjacent tissues from PTC patients were collected. Specimens were collected for immunohistochemical staining, and the results were scored and counted. The results showed that MRPL9 was highly expressed in PTC tissues (Figure 1C,D). We further divided patients into a low expression group (*n* = 13) and a high expression group (*n* = 13) according to the median expression level of MRPL9. The correlation between the expression level of MRPL9 and the clinicopathological characteristics of PTC patients, such as age, gender, TNM stage, multifocality, distant metastasis, and lymph node metastasis, is shown in Table 1. The high MRPL9 expression group was associated with a higher degree of TNM progression, extrathyroidal extension and lymph node metastasis (*p* ≤ 0.05). However, there was no significant correlation between MRPL9 level and patient’s gender and age (*p* > 0.05). The protein expression level of MRPL9 in patients’ tissues was further analyzed by western blot experiment, and western blot quantitative analysis was performed. The results showed that MRPL9 was highly expressed in PTC cancer tissues (Figure 1E, F). The above results indicated that MRPL9 was highly expressed in PTC tissues. In addition, we further explored the expression level of MRPL9 in PTC cell lines, qRT-PCR and Western blot results showed that, compared with Nthy-ori 3.1 cell lines, the mRNA and protein levels of MRPL9 were higher in TPC-1, K1 and BCPAP cell lines (Figure 1G–I).

### 2.2. MRPL9 Promotes the Proliferation and Migration of PTC Cells

In view of the high expression of MRPL9 in PTC cell lines and to explore the role of MRPL9 in PTC cell lines, K1 and BCPAP cell lines were infected with lentivirus to obtain stable cell lines that stably overexpressed or stably knocked down MRPL9. Western blot and quantitative results showed that the protein level of MRPL9 in the LV- MRPL9 group was significantly increased in K1 and BCPAP cell lines compared with the LV-Vector group (Figure 2A,B). Compared with the LV-shNC group, the protein expression level of MRPL9 in the LV-sh MRPL9 group was significantly down-regulated in K1 and BCPAP cell lines (Figure 2C,D). This indicated that we had successfully constructed cell lines stably overexpressing or stably knocking down MRPL9. The cell clone formation assay showed that K1 and BCPAP cells grew faster in the overexpression MRPL9 group. Conversely, knockdown of MRPL9 inhibited cell growth (Figure 2E,F). The results of wound healing and transwell assays showed that overexpression of MRPL9 promoted the migration ability of K1 and BCPAP cells, while knockdown of MRPL9 had the opposite function (Figure 2G–J). The above results indicated that MRPL9 promoted the proliferation and migration ability of PTC cells.

### 2.3. Interaction between GGCT and MRPL9

In order to further explore the molecular mechanism of MRPL9-mediated PTC cell proliferation and migration, starbase database (http://starbase.sysu.edu.cn/index.php, accessed on 22 March 2022) predicted that the expression levels of MRPL9 and GGCT were positively correlated (Figure 3A). In addition, the results of immunofluorescence detection showed that the localization of MRPL9 and GGCT in cells partially overlapped (Figure 3B), and the immunofluorescence homologous double-labeling experiment of PTC tissue further demonstrated the partial overlap of the intracellular localization of MRPL9 and GGCT (Figure 3C). These results suggested the possibility of an interaction between MRPL9 and GGCT. Western blot results indicated that overexpression of GGCT could promote the protein expression of MRPL9, while knockdown of GGCT inhibited the expression of MRPL9 (Figure 3D–F). To further validate the interaction between GGCT and MRPL9, Co-IP experiments were performed on K1 and BCPAP cells. Compared with the IgG group, Anti-GGCT enriched a stronger signal of MRPL9 (Figure 3G), which further proved the interaction between GGCT and MRPL9.

### 2.4. GGCT Is Highly Expressed in PTC

To explore the expression level of GGCT in PTC, 26 pairs of cancer tissues and para-cancerous tissues of PTC patients were collected. The collected samples were stained with IHC, and the IHC results were scored and counted. The results showed that GGCT was highly expressed in PTC tissues (Figure 4A,B). In addition, statistical analysis of the IHC score of GGCT found that the expressions of GGCT and MRPL9 were positively correlated (Figure 4C), which further confirmed the interaction between GGCT and MRPL9. In order to further prove the high expression of GGCT in PTC, the protein expression level of GGCT in the patients’ tissues was analyzed by western blot experiment and the quantitative analysis of western blot was performed. The results showed that GGCT was highly expressed in PTC cancer tissue (Figure 4D,E). The above results indicated that GGCT was highly expressed in PTC tissues. In addition, we further explored the expression level of GGCT in PTC cell lines, qPCR and Western blot results showed that compared with Nthy-ori 3.1 cell lines, GGCT was highly expressed in mRNA and protein levels in TPC-1, K1, and BCPAP cell lines (Figure 4F–H).

### 2.5. GGCT Regulates the MAPK/ERK Signaling Pathway by Interacting with MRPL9

Next, we explored the molecular mechanism by which GGCT and MRPL9 promotes the proliferation and migration of PTC cells. Thyroid cancers have been reported in the literature to be predominantly MAPK-driven cancers, and approximately 70% of thyroid cancers are caused by mutations that activate this pathway [27]. Further studies showed that knockdown of GGCT/MRPL9 could significantly inhibit the expression of p-P38/p-ERK, and the inhibition was more pronounced in the sh-GGCT+sh-MRPL9 group (Figure 5A–C). These results suggested that MRPL9 regulated the MAPK/ERK pathway through interaction with GGCT.

### 2.6. Knockdown of GGCT and MRPL9 Inhibited Tumor Growth and Metastasis In Vivo

In order to explore the role of GGCT/MRPL9 in vivo, nude mice were subcutaneously xenografted or injected with GGCT/MRPL9 in the tail vein to explore the effect of GGCT/MRPL9 on the proliferation and migration of PTC cells in vivo. Luciferase-labeled K1 cells were infected with lentivirus containing shGGCT/shMRPL9/shGGCT + shMRPL9, and then equal amounts of cells were injected subcutaneously into nude mice. The bioluminescence intensity of the subcutaneous tumors was detected on the Xenogen IVIS Lumina System 28 days after the subcutaneous injection and the xenografts were harvested. The results showed that the fluorescence intensity of the subcutaneous xenografts in the shGGCT/shMRPL9 group was weakened (Figure 6A,B), and the mass and volume of the subcutaneous tumors decreased (Figure 6C–E), while the fluorescence intensity, mass and volume of the xenografts in the shGGCT + shMRPL9 group were the smallest (Figure 6A–E). For the nude mouse lung metastasis model, luciferase-expressing sh-NC-K1/shGGCT-K1/shMRPL9-K1/shGGCT + shMRPL9-K1 cells were injected into the tail vein, and the bioluminescence signal in the lungs of nude mice was recorded every 7 days. With the prolongation of observation time, the fluorescent signal in the lungs of nude mice in the shGGCT/shMRPL9 group weakened compared with the control group, and the fluorescent signal in the shGGCT + shMRPL9 group was the weakest (Figure 6F,G). The nude mice were sacrificed on the 28th day after injection. From the bioluminescence signals in the living lungs of nude mice on day 28, we preliminarily concluded that knockdown of GGCT/MRPL9 attenuated the ability of K1 cells in the lung metastases of nude mice, while knockdown of GGCT and MRPL9 had the best inhibitory effect (Figure 6H). The results of further detection of the fluorescence signal of the nude mouse lung tissue in vitro were consistent with the change trend of the fluorescence intensity of the nude mice in vivo (Figure 6I). Further H&E staining of the lung tissue of nude mice showed that the number of tumor nodules in the shGGCT + shMRPL9 group was reduced compared with the control group, and the number of tumor nodules in the shGGCT + shMRPL9 group was the least (Figure 6J and Appendix A). The above results indicated that knockdown of GGCT/MRPL9 inhibited the proliferation and migration of tumor cells in vivo, and simultaneous knockdown of GGCT and MRPL9 had the strongest inhibitory effect on the proliferation and migration of PTC cells.

## 3. Discussion

In this study, GGCT and MRPL9 were shown to be involved in regulating the biological behavior of PTC tumor cells and promoting the further development of cancer, suggesting that GGCT and MRPL9 may serve as potential biomarkers. Further research found that GGCT/MRPL9 promotes the MAPK/ERK signaling pathway to promote the proliferation and migration of PTC cells. The function of GGCT/MRPL9 in PTC cells was also confirmed in in vivo experiments.

Thyroid cancer originates from follicular epithelial cells or parafollicular C cells. Follicular cell-derived thyroid cancers are classified into 4 histological types: papillary thyroid cancer (PTC 80–85%), follicular thyroid cancer (FTC 10–15%), poorly differentiated thyroid cancer (PDTC, <2%) and anaplastic thyroid cancer (ATC, <2%) [28]. PTC is primarily associated with mutations that activate the MAPK (mitogen-activated protein kinase) signaling pathway, such as RET, NTRK (neurotrophic receptor tyrosine kinase) and ALK (anaplastic lymphoma kinase) gene rearrangements or RAS (rat sarcoma) and BRAF (rapidly accelerating fibrosarcoma type B) activating point mutations, almost always found in an exclusive manner, suggesting that oncogenic activation of one member of this pathway was sufficient to drive transformation [29,30,31]. Furthermore, Xing M demonstrated that activation of the MAPK signaling pathway resulted in the up-regulation of tumor-promoting genes (such as VEGFA, MET, HIF1A, UPA, UPAR, TGFB1 and TSP1) and the down-regulation of tumor suppressor and thyroid genes (such as TIMP3, SLC5A8, DAPK1, NIS, TSHR and TPO) [32]. Our study found that functional inhibition of MRPL9/GGCT would inhibit the activity of the MAPK signaling pathway, and, simultaneously, inhibiting the protein expression of MRPL9 and GGCT would aggravate the inhibition of the MAPK signaling pathway. This indicated that MRPL9/GGCT promoted the occurrence and deterioration of PTC by activating the activity of MAPK signaling pathway.

Dysregulation of mitochondrial bioenergetics has been recognized as a key metabolic hallmark of malignancy [33]. In recent years, some MRPs have been found to be involved in multiple cellular processes, such as cell proliferation, ribosome cycle regulation in vitro, and apoptosis, and the expression level of MRPL20 was significantly down-regulated in androgen-independent prostate cancer [34,35,36]. The levels of MRPL37 and mRNA were also significantly elevated in different lymphoma tissues. MRPL33 is required for mitochondrial function and has been implicated in tumor progression [34,36,37]. Recently, PTCD3 (also known as MRPS39) was shown to be critical for the maintenance of Myc-driven lymphomas [38]. MRPL44 expression was identified as a predictor of lymph node metastasis in papillary thyroid carcinoma [39], and MRPL13 inhibition was shown to be a key upstream regulator of OXPHOS dysfunction and hepatoma cell invasiveness [40]. These results suggest that dysregulation of MRPs may be responsible for bioenergetic dysregulation and play a key role in tumor progression. MRPL9 has been reported to be highly expressed in HCC and associated with poor prognosis [12,13], and knockdown of MRPL9 was determined to significantly inhibit cell proliferation and migration in HCC [13]. Our study found that MRPL9 was highly expressed in PTC, and MRPL9 promoted the proliferation and migration ability of PTC cells, which was further proved in the nude mouse subcutaneous xenograft model and lung metastasis model. Further studies revealed the interaction between MRPL9 and GGCT, which activated the MAPK/ERK signaling pathway and further promoted the malignancy of PTC.

Gamma-glutamyl cyclotransferase (GGCT, 188 amino acids, 21 kDa) is an enzyme involved in the metabolism of glutathione, which catalyzes the conversion of γ-Glu-AA to pyroglutamate [15]. GGCT has been reported to be upregulated in several cancers, and depletion of GGCT can exert anticancer effects in these cancers, including prostate, esophageal squamous cell carcinoma, breast, gastric, and ovarian cancers [23,41,42,43,44]. GGCT was shown to promote colorectal cancer migration and invasion through epithelial-mesenchymal transition [22]. GGCT depletion in gastric cancer significantly inhibited cell proliferation and colony-forming capacity in MGC80-3 and AGS cells, induced apoptosis in early and late gastric cancer cells and induced gastric cancer cell cycle arrest in G2/M phase [23]. A study identified inhibition of cancer cell proliferation by disrupting GGCT, highlighting the potential to treat types of malignancies by inhibiting GGCT [45]. Our study found that GGCT was highly expressed in PTC, and the interaction between GGCT and MRPL9 jointly affected the MAPK/ERK signaling pathway. Our previous study showed that GGCT knockdown reduced the proliferation, migration and invasion abilities of PTC cells in vitro and blocked the EMT process. Mechanistically, it was found that GGCT can bind to CD44 and that GGCT plays a role in stabilizing CD44 to prevent its degradation [26]. Whether the interaction between GGCT and MRPL9 in this study is based on the modification of MRPL9 by GGCT remains to be further investigated.

In conclusion, our study shows that MRPL9/GGCT acts as an oncogene in PTC to promote the malignant progression of PTC by activating the MAPK/ERK signaling pathway. The findings provide new insights into the diagnosis and treatment of PTC.

## 4. Materials and Methods

### 4.1. Clinical Samples

The cancer tissues and adjacent normal tissues (*n* = 26) of PTC patients involved in this study were provided by Tongji Hospital of Huazhong University of Science and Technology. All participants gave informed consent and did not receive radiotherapy or chemotherapy before surgery. The research protocol was approved by the Ethics Committee of Tongji Hospital, Huazhong University of Science and Technology.

### 4.2. Cell Culture

Human normal thyroid cell line Nthy-ori 3-1; PTC cell line TPC-1, BCPAP and K1; human embryonic kidney 293T (HEK293T) cell line was purchased from Shanghai Cell Bank, Chinese Academy of Sciences. Nthy-ori 3-1, TPC-1, BCPAP cell lines were reared with RPMI-1640 complete medium (Meilunbio, Dalian, China), while 293T and K1 were reared in DMEM-H complete medium (Meilunbio, Dalian, China). All media contained 10% fetal bovine serum (FBS) (ExCell Bio, Shanghai, China) and 1% penicillin-streptomycin (Beyotime, Shanghai, China), and the cell culture conditions were 37 °C, 5% carbon dioxide.

### 4.3. Plasmid Construction and Lentivirus Assay

For the construction of the expression plasmid for GGCT/MRPL9, the GGCT/MRPL9DE CDS sequence was amplified from human cDNA and ligated into the pLVX-puro (Addgene, Watertown, MA, USA) vector. The procedure followed the instructions for the cloning kit and EcoRI restriction endonuclease sites were used (10911ES20, Yeasen, Shanghai, China). The plasmid of shGGCT/shMRPL9 was synthesized in Tsingke Biotechnology Co., Ltd. (Tsingke Biotechnology, Beijing, China). The PCR amplification primers and sh-RNA sequences mentioned above are provided in Appendix A. For lentivirus packaging, PEI (40820ES04, Yeasen, Shanghai, China). was used to transfect the target plasmid, pCMV-VSV-G (Beyotime, Shanghai, China) and pCAG-dR8.9 (Beyotime, Shanghai, China) in 293T cells (according to the ratio of target plasmid: pCMV-VSV-G: pCAG-dR8.9 = 4:3:1). The medium was replaced with fresh medium 6 h after transfection, and the virus stock solution was collected 48 h after transfection. For stably transfected cell lines, PTC cells were infected with LV-GGCT, LV-MRPL9, LV-sh-GGCT, LV-sh-MRPL9, LV-pLVX, LV-pLKO.1, then cells were treated with 1 ug/mL puromycin (Meilunbio, Dalian, China).

### 4.4. Quantitative Real-Time PCR(qRT-PCR) Assay

Total RNA was extracted with Trizol (R0016, Beyotime, Shanghai, China), 1 µg RNA was reverse transcribed into cDNA by reverse transcription kit (R212, Vzayme, Nanjing, China), and qPCR reaction was performed with SYBR Green (11201ES03, Yeasen, Shanghai, China). The qRT-PCR program was set as follows: 95 °C for 5 min, followed by 40 cycles of 95 °C for 10 s, 60 °C for 20 s, and 72 °C for 20 s (CFX96, Bio-Rad, Hercules, CA, USA). Actin was used as normalization and the relative levels of gene expression were determined by the 2^−ΔΔCT^ method. Primer information is provided in Appendix A.

### 4.5. Western Blot Assay

Total protein from cells or patient tissue was extracted using RIPA lysis buffer (Beyotime, Shanghai, China), and an equal amount of protein was electrophoresed on a polyacrylamide gel and then transferred to a polyvinylidene fluoride (PVDF) membrane (Millipore, Burlington, MA, USA). The obtained PVDF membrane was incubated with 5% BSA at room temperature for 1 h, and then the PVDF membrane was incubated with the primary antibody and the secondary antibody successively, and developed on the imaging system (Bio-Rad, Hercules, CA, USA) after adding the highly sensitive ECL reagent. The experimental results were quantitatively analyzed with Image J. The antibody information used in the study is as follows: Anti-GGCT (1:1000, Proteintech, 16257-1-AP, Wuhan, China); Anti-MRPL9 (1:1000, Proteintech, 15342-1-AP, Wuhan, China), Anti-Actin (1:5000, ABclonal, AC026, Wuhan, China); anti-P-p38 (1:1000, CST, 29216, Danvers, MA, USA), anti-p38 (1:1000, ABclonal, A5049, Wuhan, China), anti-ERK1/2 (1:1000, CST, 4695, Danvers, MA, USA), anti-p-ERK1/2 (1:1000, CST, 4370, Danvers, MA, USA).

### 4.6. Transwell Assay

The transwell chambers (Corning, 3422, Corning, NY, USA) were coated with Matrigel and placed in 24-well plates to incubate overnight with culture medium. The treated cells were resuspended in FBS-free medium, and 2 × 10^4^ cells/well were seeded into the upper chamber. A medium containing 10% FBS was then added to the lower chamber and incubated for 24 h. After removal of non-migrating cells, the chambers were fixed in 4% paraformaldehyde (MA0192, Meilunbio, Dalian, China) for 30 min and stained with 1% crystal violet (MA0148, Meilunbio, Dalian, China) for 30 min, and images were acquired using a microscope.

### 4.7. Wound Healing Assay

A total of 1 × 10^5^ treated cells were seeded into each well of a 6-well plate and cultured to 90% confluence. Then, a 200 µL pipette tip was used to create a wound in the middle of each well. After washing with PBS, cells were incubated with medium containing 1% FBS. Wound images were acquired at 0 and 36 h. Each experiment was performed in triplicate.

### 4.8. Colony Formation Assay

For cell colony formation experiments, treated cells were seeded into 6-well plates (5 × 10^3^). After 12 days of incubation at 37 °C in RPMI-1640/DMEM-H medium containing 10% FBS, the plates were washed with PBS, and stained with 0.1% crystal violet for 30 min at room temperature. Each experiment was repeated ≥3 times.

### 4.9. Immunofluorescence Assay

For immunofluorescence experiments, briefly, K1/BCPAP cells were seeded into 24-well plates (1 × 10^4^/well) containing cell slides, where K1/BCPAP cells stably overexpressed flag-tagged GGCT. After cells adhered, cells were fixed with 4% paraformaldehyde (MA0192, Meilunbio, Dalian, China), permeabilized with 0.1% Triton X-100 (P0096, Beyotime, Shanghai, China), blocked with 5% BSA, and incubated with primary antibody and fluorescent secondary antibody, respectively. Finally, the nuclei were stained with DAPI and observed under a confocal microscope. Information on the primary and secondary antibodies used is as follows: MRPL9 (1:1000, Proteintech, 15342-1-AP, China), anti-flag-tag (1:50, ABclonal, AE005, China), FITC-labeled goat anti-mouse IgG (1:100, Servicebio, GB22301, Wuhan, China), Cy3-labeled goat anti-rabbit IgG (1:100, Servicebio, GB21303, China).

### 4.10. Co-Immunoprecipitation (Co-IP) Assay

For Co-IP assay, whole cell lysates were obtained by Western and IP cell lysate (Beyotime, Shanghai, China) lysis buffer and incubated overnight with anti-GGCT (1:100, Proteintech, 16257-1-AP, Wuhan, China) or anti-IgG (1:100, ABclonal, AC005, Wuhan, China) at 4 °C. It was then incubated with rProtein A/G Plus MagPoly Beads (RM09008, ABclonal, Wuhan, China) for 2 h at 4 °C. Finally, the protein bound to the beads was eluted and detected by western blotting using antibodies against GGCT (1:1000, Proteintech, 16257-1-AP, Wuhan, China) or MRPL9 (1:1000, Proteintech, 15342-1-AP, Wuhan, China).

### 4.11. Mouse Model

For the mouse xenograft model, 16 BALB/c nude mice (four weeks old, female) were randomly divided into four groups: K1-shNC, K1-shGGCT, K1-shMRPL9, shGGCT + shMRPL9. Luciferase-labeled K1 cells transduced with K1-shNC, K1-shGGCT, K1-shMRPL9, shGGCT + shMRPL9 viruses were harvested, resuspended in PBS, and then inoculated subcutaneously into nude mice (1 × 10^7^). Twenty-eight days later, the subcutaneous tumors were harvested for subsequent analysis after imaging with the Xenogen IVIS Lumina System (Caliper Life Sciences, Waltham, MA, USA).

For the lung metastasis mouse model, the same grouping was used as the mouse xenograft model. Luciferase-labeled K1 cells transduced with K1-shNC, K1-shGGCT, K1-shMRPL9, shGGCT + shMRPL9 were harvested and resuspended in PBS, and injected into nude mice via tail vein (1 × 10^6^). Bioluminescence imaging (BLI) was used to monitor the growth of lung K1-fLuc metastases every 7 days. After 28 days, the lung tissues of nude mice were harvested for subsequent analysis. All animal protocols were approved by the Laboratory Animal Ethics Committee of Wuhan University of Science and Technology. This research was conducted according to the principles of the Declaration of Helsinki.

### 4.12. Hematoxylin-Eosin (H&E) Staining and Immunohistochemistry (IHC) Assay

Tissues were sent to Sevier Bio (Wuhan, China) for embedding and sectioning. For H&E staining, sections were deparaffinized and then stained according to the instructions of the hematoxylin-Eosin (H&E) staining kit (E607318, Sangon Biotech, Shanghai, China). For immunohistochemistry, after the paraffin sections were dewaxed and rehydrated, the tissue sections were placed in a repair box filled with citric acid antigen retrieval buffer (PH 6.0), heated in a microwave oven for antigen retrieval, and then blocked endogenous peroxidase activity with 3% hydrogen peroxide solution. After incubation with specific primary and secondary antibodies, color is developed by DAB color development kit (G1212, Servicebio, Wuhan, China). The required antibody information is as follows: GGCT (1:100, Proteintech, 16257-1-AP, Wuhan, China), MRPL9 (1:100, Proteintech, 15342-1-AP, Wuhan, China). The immunohistochemical results were evaluated as follows: staining intensity was divided into 3 grades (0 = no staining, 1 = weak staining, 2 = moderate staining, 3 = strong staining), and the percentage of positive areas was divided into 4 grades (0% (0), <10% (1), 10–30% (2), 31–70% (3), 71–100% (4)). The final score for immunohistochemistry was determined by multiplying the staining intensity score by the percent positive area score, up to a maximum of 12 points. Proteins with high expression (≥6) and low expression (<6) were based on the final score. The protein expression of MRPL9 and GGCT (score) was assessed by two independent pathologists.

### 4.13. Paraffin Section Immunofluorescence Homologous Double Labeling Assay

Briefly, paraffin sections were dewaxed and the sections were placed in a retrieval box containing EDTA antigen retrieval buffer (pH 8.0) (G1206, Servicebio, Wuhan, China) for antigen retrieval in a microwave oven. Then, the sections were placed in a 3% hydrogen peroxide solution to block endogenous peroxidase. After blocking with 3% BSA, add Anti-GGCT and incubate overnight at 4 degrees Celsius. After incubation with secondary antibody, add FITC-TSA (G1222, Servicebio, Wuhan, China) and incubate in the dark for 10 min. Next, the tissue sections were placed in a repair box filled with EDTA antigen retrieval buffer (PH8.0) and heated in a microwave oven to remove the primary and secondary antibodies that had been bound to the tissue, and then incubated with Anti-MRPL9 and fluorescent secondary antibodies (1:100, GB21303, Servicebio, Wuhan, China) respectively. After counterstaining the nuclei with DAPI, an autofluorescence quencher (G1221, Servicebio, Wuhan, China) was added, and finally the slides were mounted.

### 4.14. Bionformatic Analysis

The TCGA database (https://www.cancer.gov/aboutnci/organization/ccg/research/structural-genomics/tcga, accessed on 14 March 2022) analyzed the expression level of MRPL9 in thyroid cancer, and the GEPIA 2 (http://gepia2.cancer-pku.cn/#index, accessed on 14 March 2022) website predicted the survival and prognosis of MRPL9 in thyroid cancer. In addition, the Starbase database website (http://starbase.sysu.edu.cn/index.php, accessed on 5 April 2022) predicted the expression correlation between MRPL9 and GGCT.

### 4.15. Statistical Analysis

Statistical analysis was performed using SPSS 19.0 software (SPSS Inc., Chicago, IL, USA) and GraphPad Prism 8. The measurement data (mean ± standard deviation) between the two groups were compared by paired *t*-test, and multiple groups were compared by one-way analysis of variance (ANOVA). *p* < 0.05 was considered statistically significant.

## Figures and Tables

**Figure 1 ijms-23-11989-f001:**
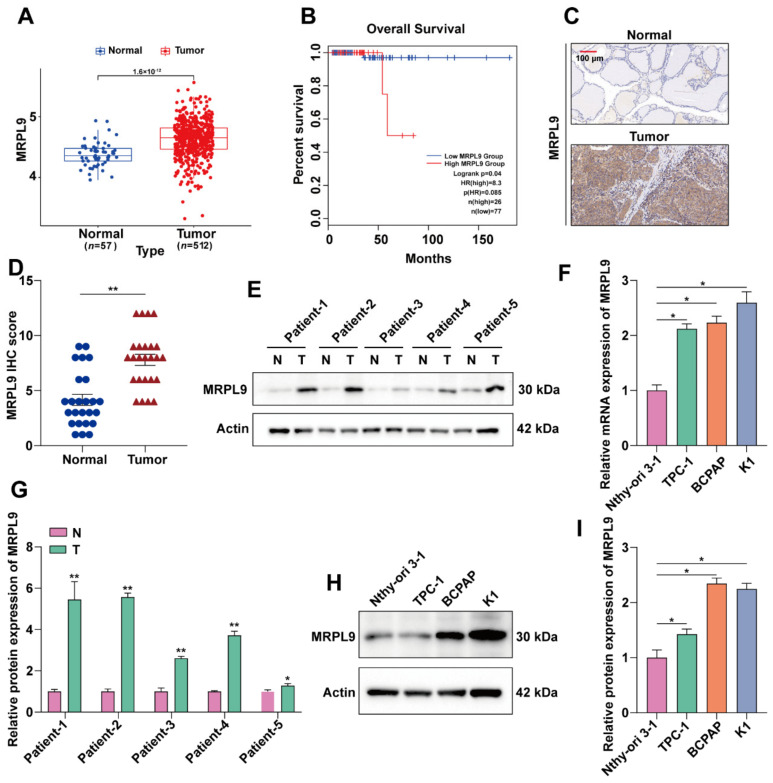
MRPL9 is highly expressed in PTC. (**A**) The expression level of MRPL9 in thyroid cancer was analyzed by TCGA database. (**B**) The GEPIA2 website (http://gepia2.cancer-pku.cn/#index, accessed on 22 March 2022) analyzed the effect of MRPL9 on the survival and prognosis of patients with thyroid cancer. (**C**) Representative images of MRPL9 immunohistochemistry (IHC) of normal PTC tissue (top) and PTC tissue (bottom) (IHC score: 10, scar bar = 100 µm). (**D**) MRPL9 immunohistochemical score statistics of PTC group (*n* = 26) and normal group (*n* = 26) (Each dot represents a patient’s normal thyroid tissue, and each triangle represents a patient’s PTC tissue). (**E**) Representative Western blot of MRPL9 protein in 26 pairs of PTC tissues and normal tissues. (**F**) Western blot results were quantified and normalized to Actin (*n* = 3). (**G**) RT-qPCR analysis examined the expression level of MRPL9 in Nthy-ori 3.1, TPC-1, K1, BCPAP cell lines (*n* = 3). (**H**) Western blot detection of MRPL9 protein expression levels in Nthy-ori 3.1, TPC-1, K1, and BCPAP cell lines. (**I**) Western blot results were quantified and normalized to Actin (*n* = 3). Measurement data (mean ± standard deviation) between two groups were compared by paired *t* test, and multiple groups were compared by one-way analysis of variance (ANOVA) with Tukey’s post hoc test. “*” represents *p* ≤ 0.05, “**” represents *p* ≤ 0.01.

**Figure 2 ijms-23-11989-f002:**
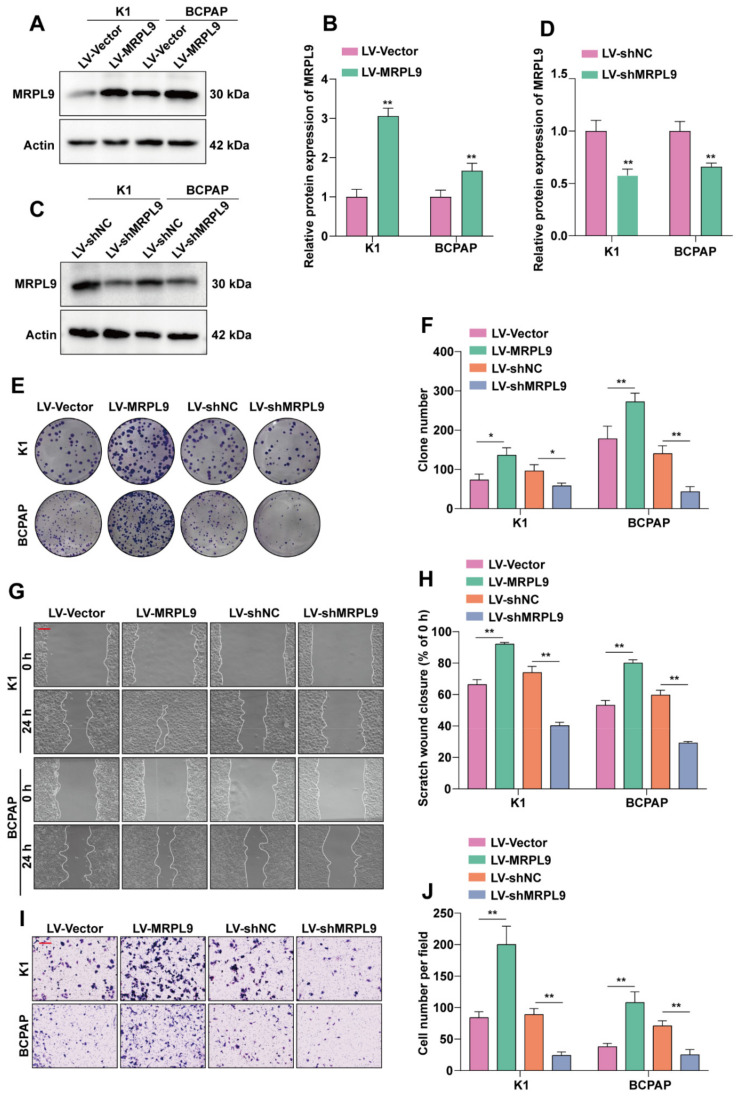
MRPL9 promotes the proliferation and migration of PTC cells. (**A**) K1 and BCPAP cell lines were infected with lentivirus containing LV-Vector, LV-MRPL9, and the overexpression efficiency of MRPL9 was detected by Western blot. (**B**) Western blot results were quantified and normalized to Actin (*n* = 3). (**C**) K1 and BCPAP cell lines were infected with lentivirus containing LV-shNC and LV-shMRPL9, and the knockdown efficiency of MRPL9 was detected by Western blot. (**D**) Western blot results were quantified and normalized to Actin (*n* = 3). (**E**) Colony formation assays assessed the effect of MRPL9 overexpression or knockdown on the proliferative capacity of K1 and BCPAP cells. (**F**) Statistical analysis of the number of cell clones was performed in image J (*n* = 3). (**G**) The effect of MRPL9 overexpression or knockdown on the migration ability of K1 and BCPAP cells was examined by wound healing assay (scar bar = 100 µm). (**H**) Statistical analysis of cell wound area was performed with image J (*n* = 3). (**I**) Transwell assay to detect the effect of MRPL9 overexpression or knockdown on the migration ability of K1 and BCPAP cells (scar bar = 100 µm). (**J**) Statistical analysis of cell numbers was performed with image J (*n* = 3). Measurement data (mean ± standard deviation) between two groups were compared by paired *t* test. “*” represents *p* ≤ 0.05, “**” represents *p* ≤ 0.01.

**Figure 3 ijms-23-11989-f003:**
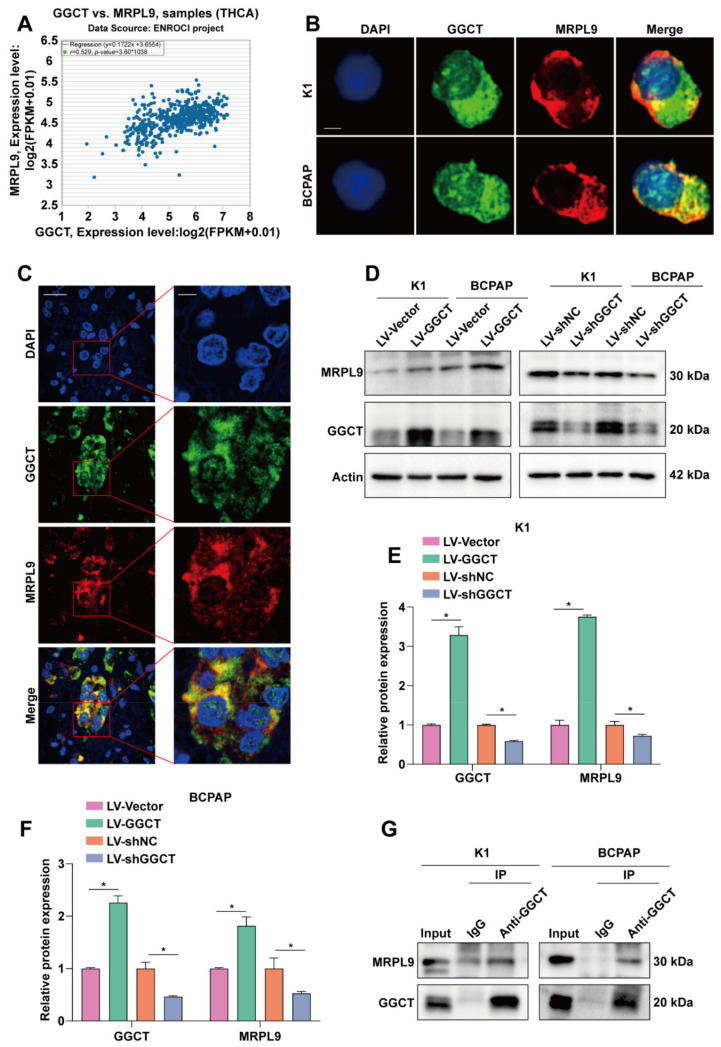
Interaction between GGCT and MRPL9. (**A**) The Starbase database (http://starbase.sysu.edu.cn/index.php, accessed on 22 March 2022) analyzed the correlation between GGCT and MRPL9 in thyroid cancer. (**B**) Immunofluorescence assays explored the localization of MRPL9 and GGCT in cells (scar bar = 5 µm). (**C**) Representative image of the localization of MRPL9 and GGCT detected by immunofluorescence homologous double labeling experiments on paraffin sections (Left figure, scar bar = 20 µm; Right figure, scar bar = 5 µm). (**D**) GGCT was overexpressed/knocked down in K1/BCPAP cells, and the protein expression level of MRPL9 was detected by Western blot. (**E**) The expression levels of GGCT and MRPL9 in the K1 cell line were analyzed with image J, and normalized to Actin (*n* = 3). (**F**) The expression levels of GGCT and MRPL9 in the BCPAP cell line were analyzed with image J, and normalized to Actin (*n* = 3). (**G**) The interaction between GGCT and MRPL9 was analyzed by Co-IP in K1/BCPAP cells. Measurement data (mean ± standard deviation) between two groups were compared by paired *t* test. “*” represents *p* ≤ 0.05.

**Figure 4 ijms-23-11989-f004:**
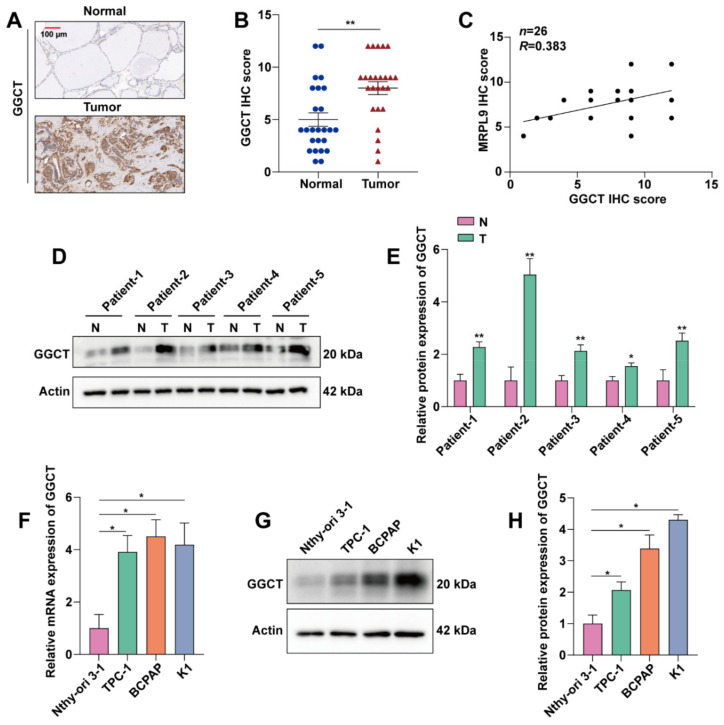
GGCT is highly expressed in PTC. (**A**) Representative images of GGCT immunohistochemistry (IHC) of normal PTC tissue (left) and PTC tissue (right) (scar bar = 100 µm). (**B**) Statistics of GGCT immunohistochemical score of PTC group (*n* = 26) and normal group (*n* = 26) (Each dot represents a patient’s normal thyroid tissue, and each triangle represents a patient’s PTC tissue). (**C**) Correlation analysis of GGCT and MRPL9 IHC scores. (**D**) Representative Western blot of GGCT protein in 26 pairs of PTC tissues and normal tissues. (**E**) Western blot results were quantified and normalized to Actin (*n* = 3). (**F**) RT-qPCR analysis detected the expression level of GGCT in Nthy-ori 3.1, TPC-1, K1, BCPAP cell lines (*n* = 3). (**G**) Western blot detection of GGCT protein expression levels in Nthy-ori 3.1, TPC-1, K1, and BCPAP cell lines. (**H**) Western blot results were quantified and normalized to Actin (*n* = 3). Measurement data (mean ± standard deviation) between two groups were compared by paired *t* test, and multiple groups were compared by one-way analysis of variance (ANOVA) with Tukey’s post hoc test. “*” represents *p* ≤ 0.05, “**” represents *p* ≤ 0.01.

**Figure 5 ijms-23-11989-f005:**
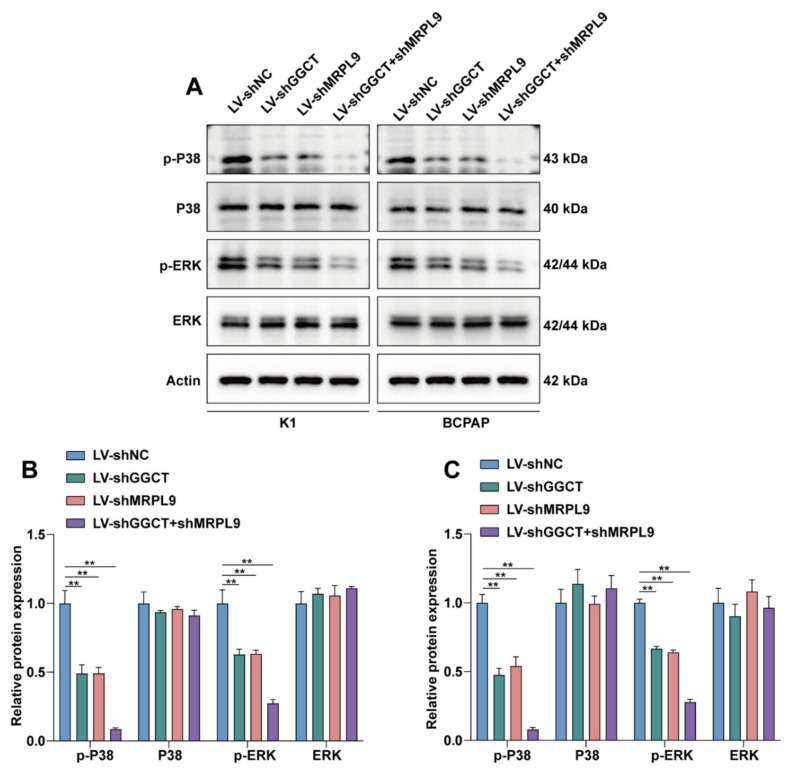
GGCT regulates the MAPK/ERK signaling pathway by interacting with MRPL9. (**A**). The protein expression levels of P38, p-P38, ERK and p-ERK in the shGGCT/shMRPL9/sh-GGCT + shMRPL9 group were detected by Western blot in K1/BCPAP cells. (**B**) The expression levels of P38, p-P38, ERK and p-ERK in the shGGCT/shMRPL9/sh-GGCT + shMRPL9 group in K1 cell line were analyzed by J image, normalized to Actin (*n* = 3). (**C**) The expression levels of P38, p-P38, ERK and p-ERK in the shGGCT/shMRPL9/sh-GGCT + shMRPL9 group in BCPAP cell line were analyzed by J image, normalized to Actin (*n* = 3) Measurement data (mean ± standard deviation) between two groups were compared by paired *t* test. “**” represents *p* ≤ 0.01.

**Figure 6 ijms-23-11989-f006:**
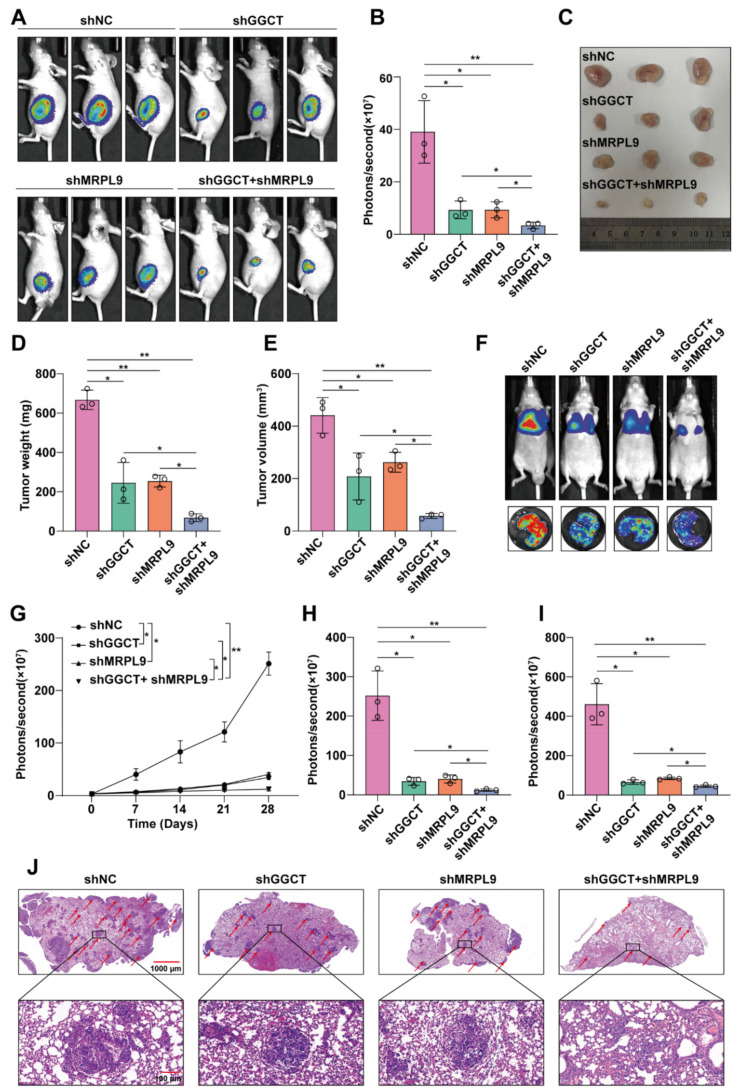
Knockdown of GGCT and MRPL9 inhibited tumor growth and metastasis in vivo. (**A**,**B**) The Xenogen IVIS Lumina System detects the bioluminescence intensity of subcutaneous xenografts in nude mice in vivo, and performs statistical analysis on the fluorescence signals (*n* = 3). (**C**) Subcutaneous xenografts in nude mice. (**D**,**E**) Statistical analysis of mass and volume of subcutaneous xenografts (*n* = 3). (**F**) The Xenogen IVIS Lumina System detects bioluminescence intensity in the lungs of nude mice in vivo (top) and ex vivo (bottom) (*n* = 3). (**G**) The line chart denotes luciferase bioluminescence emitted from the lungs in each group over time (0–28 days). (**H**) Statistical analysis of bioluminescence intensity in the lungs of nude mice in vivo (*n* = 3). (**I**) Statistical analysis results of bioluminescence intensity of nude mouse lung tissue in vitro (*n* = 3). (**J**) HE staining was used to analyze the metastases in the lungs of nude mice. Measurement data (mean ± standard deviation) between two groups were compared by paired *t* test, and multiple groups were compared by one-way analysis of variance (ANOVA). “**” represents *p* ≤ 0.01, “*” represents *p* ≤ 0.05.

**Table 1 ijms-23-11989-t001:** Relationship between MRPL9 expression and clinicopathological characteristics in 26 PTC tissues.

Clinicopathologic Parameters	n	GGCT Expression		*p*
		Low	High	
All cases	26	13	13	
Age				0.724
≤55	14	6	8	
>55	12	7	5	
Gender				0.342
male	10	6	4	
female	16	7	9	
Multifocality				0.821
unifocal	12	6	6	
multifocal	14	7	7	
Extrathyroidal extension				0.023
no	15	10	5	
yes	11	3	8	
Lymph node metastasis				0.041
no	12	8	4	
yes	14	5	9	
T classification				0.009
T1-T2	14	10	4	
T3-T4	12	3	9	
TNM stage				0.005
I + II	12	9	3	
III + IV	14	4	10	

## Data Availability

The datasets used or analysed during the current study are available from the corresponding author on reasonable request.

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
