# Peer review of "Interaction of MRPL9 and GGCT Promotes Cell Proliferation and Migration by Activating the MAPK/ERK Pathway in Papillary Thyroid Cancer"

_ijms, 2022, doi:10.3390/ijms231911989_

Round 1

Reviewer 1 Report

Zhang et al. showed the involvement of MRPL9 and GGCT in papillary thyroid cancer. The authors analyzed data from databases, PTS tissue, performed in vitro and in vivo experiments. They found that both proteins are highly expressed in PTS tissues and cell lines and their knockdown inhibited tumor growth by diminished MAPK/ERK pathway.

Major concerns

Results

2.1: You analyzed data of TCGA database. Can you include information about impact of MRPL9 on patient survival and prognosis (beside expression data)?

Please introduce the different cell lines (ncy-ori 3.1 [text]/Nthy-ori 3-1 [Fig. 1], TPC-1, K1, BCPAP)! All cell lines with PTS characteristics (human)? You can include information in materials and methods section or results when firstly mentioned.

2.3: In my opinion, these experiments didn´t show a real interaction of MRPL9 and GGCT. If you compare expression of MRPL9 (Fig. 1B) and GGCT (Fig. 4A) of patient samples, the distribution looks different. To support your thesis of co-expression, you can stain both proteins together on patient samples to define their localization (or in serial sections for IHC).

Fig. 3B: For confocal microscopy, this resolution is not enough to define co-localisation of proteins. A higher magnification is required. Additionally, you have to include a marker for cell structure to define localization of proteins within the cell. It looks like almost 100% co-expression. Did you perform negative controls, or check, if antibodies didn´t show interaction with each other? These pictures didn´t proof that MRPL9 and GGCT interact.

Fig. 3F: IP isn´t convincing. With IgG control you got also MRPL9 protein. How is the difference between IgG control and anti-GGCT? Or explain/discuss protein bands.

Fig. 4C: you wrote “n=25”, but only 17 data points are shown

2.5: L187/188: overstatement “These results suggest that GGCT regulates the MAPK/ERK pathway by directly binding to MRPL9.” Direct binding is not proven by experiments. Please rephrase or include better experiments for showing direct binding of these both proteins.

Fig. 6: Using only 3 animals per experimental group is atypical. Which statistical test did you use for calculating significances?

Materials and Methods

4.1: Please include patient characteristics!

Description of bioinformatic analysis is missing!

4.3: A detailed description of the cloning and transfection is required.

4.4: PCR program, product size, device for RT-PCR.

4.5: Include blocking strategy

4.9: Include used primary and secondary antibodies (name, company, dilution)

4.10. Did you use the same antibodies as in 4.5?

4.12: How did you perform antigen retrieval? Which antibodies did you use? How many investigators analyzed the tissues and scored the staining? Blinded?

4.13: Which statistical tests did you use? Please indicate in the caption of each figure!

Minor concerns

Title

In the abstract and main text, results of MRPL9 are shown first. Therefore, think about, if in the title you can mention MRPL9 first and then GGCT.

Abstract

L15: Why did you write that mortality increased? It isn’t mentioned later.

L22: “opposite function” – better: “opposite effect”

L23: “GGC” – you mean GGCT? – please define abbreviation

L27: “findings found” – rephrase

Introduction

L57: write out in full GGCT

Figures

Think about depiction of single data points for graphs if applicable.

Fig. 1B: indicate MRPL9 IHC score for the shown example

Fig. 2G/I: indicate scale bar size; 2G: include time point for scratch analysis in caption

Fig. 5C: asterisks are missing in some places

Fig. 6C: better image or contrast enhancement is required

Fig. 6G: If you image the bioluminescence every 7 days as mentioned in the M+M. Can you provide a time course of metastasis development?

Fig. 6I: How many animals did you analyze for nodules? Are these only exemplary pictures? Or did you can also analyze tumor nodules for each group and show a graph?

Reviewer 2 Report

In the paper "Interaction of GGCT and MRPL9 Promotes Cell Proliferation 2 and Migration by Activating the MAPK/ERK Pathway in Pa- 3 pillary Thyroid Cancer”,  Zhang, H.-met al.,  provides new information on the role of MRPL9, the mitochondrial protein, and its interactions with  GGCT in papillary thyroid cancer (PTC). In particular, they showed that this interaction  of modulates the MAPK/ERK pathway, affecting the proliferation and migration of PTC cells.

Although  the results are interesting and the data analysis appear to have been competently performed, the following suggestions/comments will be fulfilled to improve the novelty and the clarity of this study:

Minor revisions

Results:

Page 3 line 94: please correct  ncy-ori 3.1 with the exact name.

Figure 1 and 4: please add the stars in each graphs

Materials and methods:

Clinical samples

Page 13 line 292: miss all clinic pathological characteristics of 25 PTC tissues used. Please add.

Cell culture

Page 13 line 299: authors mentioned HEK293 cell lines, but any results are refer to this cell line. Please fix it.

Major revisions

Discussion

As reported by authors, this present work is based on the published data (Han-Ning Li et al, Endocrinology, 2022) about the role of GGCT in PTC tumor using the same in vitro model. In the discussion authors should  add some comments regarding previous work. In additions, I am curious to know if there are some associations between MRPL9/ GGCT interaction effect and molecular alterations  which characterize the different PTC-derived cell lines. 

Round 2

Reviewer 1 Report

Now the manuscript is improved. Thank you!

Reviewer 2 Report

The authors have addressed my suggestions. The paper is accepted in its present form.